# Sexual Desire and Erotic Fantasies Questionnaire: The Development and Validation of the Erotic Fantasy Use Scale (SDEF2) on Experience, Attitudes, and Sharing Issues

**DOI:** 10.3390/healthcare11081159

**Published:** 2023-04-18

**Authors:** Filippo Maria Nimbi, Roberta Galizia, Erika Limoncin, Tom Levy, Emmanuele Angelo Jannini, Chiara Simonelli, Renata Tambelli

**Affiliations:** 1Department of Dynamic and Clinical Psychology and Health Studies, Sapienza University of Rome, 00185 Rome, Italy; roberta.galizia@uniroma1.it (R.G.); erika.limoncin@uniroma1.it (E.L.); levy.1943179@studenti.uniroma1.it (T.L.); chiara.simonelli@uniroma1.it (C.S.); renata.tambelli@uniroma1.it (R.T.); 2Department of Systems Medicine, University of Rome Tor Vergata, 00100 Rome, Italy; eajannini@gmail.com

**Keywords:** erotic fantasies, sexual desire, attitudes, psychometric properties, validation

## Abstract

Background: The investigation of sexual fantasies is a delicate issue within sex research. Most studies have focused on the content of these fantasies, rather than on use, experiences, attitudes, and sharing issues, which are fundamental aspects within sexual therapy. The main aim of the present study was to develop and validate the “Sexual Desire and Erotic Fantasies questionnaire-Part 2. Use of Erotic Fantasies (SDEF2)”. Methods: The SDEF2 was completed by 1773 Italian participants (1105 women, 645 men, and 23 other genders). Results: The final 21-item version presented a five-factor structure (fantasies frequency, fantasies normality, fantasies importance, negative emotions, and sharing and experiencing). The SDEF2 showed good psychometric properties, internal reliability, construct, and discriminant validity, appearing to be able to differentiate between sexually clinical and functional women and men (based on the FSFI and IIEF cut-off scores). Conclusions: The possibility of assessing fantasies frequency, attitudes, and emotions may be extremely useful both for research and clinical purposes. The current study seems to validate that the SDEF2 is a useful measure of assessing the different aspects related to a fantasizing activity, which was shown to be associated with sexual functioning and satisfaction.

## 1. Introduction

Erotic fantasies are considered to be among the most common human sexual experiences [1,2]. They are defined as mental imagery and thoughts that are sexually arousing or erotic to the individual while awake, and thus are not externally observable [3]. Across studies, about 90–97% of the general population report having sexual fantasies and using them to stimulate their desire or intensify their arousal [3,4,5,6,7,8].

The use of erotic fantasies is typically referred to as a positive experience [2,3,7] that is often able to activate and increase the sexual response, pleasure, and satisfaction [9,10,11,12]. Fantasizing may also hinder the effects of the negative cognitions and distracting thoughts that are commonly experienced in sexual problems [13,14,15,16,17]. In intimate relationships, sharing sexual fantasies between partners seems to increase the positive perception of the relationship, which, in turn, may motivate the partners to invest further in the relationship [12]. However, sexual fantasies can also represent a negative experience, especially when they involve non-consensual sexual activities with harmful/painful scenarios or illicit behaviors that are perceived as unwanted and distressful for the individual [6,18].

Sexual fantasies are not necessarily desires that people want to perform in real life, but they are better represented as an expression of imaginative and phantasmatic activity [6,19]. For example, having erotic fantasies related to paraphilic topics is neither rare, nor directly connected to committing a crime [7,20]. In any case, since distress is a fundamental issue in clinical work, it is necessary to evaluate not only the content, but also the frequency, emotional reactions, and attitudes towards fantasies, in order to be more able to use them as effective tools for improving sexual satisfaction and sexual health.

The literature on sexual fantasies has primarily focused on variables associated with their frequency and contents [3,11]. For example, Fisher and colleagues [21] suggested that, although there may be a gender-based difference in sexual cognitions indicating that men have more sexual fantasies than women, this difference is smaller than generally thought. Moreover, they highlight how reporting and sharing erotic fantasies with others is likely influenced by gender role expectations, social desirability, and erotophilia (i.e., the disposition to respond to sexual cues in a positive way) [22]. Sexual fantasies frequency may be affected by biological and social facets. Dawson et al. [23] showed that the frequency and arousability of sexual fantasies in women may significantly change during the menstrual cycle, increasing at ovulation. Cascalheira and colleagues [24] reported some interesting data related to the social lockdown due to the COVID-19 pandemic in the UK. They found that 34.3% of the participants in their study engaged in more sexual fantasizing during the lockdown, whereas women were more likely to report this increase than men. Living with children was a predictor of increased fantasizing. An increase in solitary sexual practices was associated with sexual fantasizing and pornography consumption. Most of the participants attributed their increases to boredom, having more free time, and replacing partnered sex.

From a methodological point of view, erotic fantasies are difficult to measure, as all attempts rely solely on self-reports, which are extremely influenced by social desirability and other possible biases [25]. These measurement challenges, together with the representability of the samples and the reliability of the measures, have contributed to the debate on how to best assess sexual fantasies [26].

Besides assessing the contents of erotic fantasies, which is not the goal of the current paper, few measures have focused on sexual fantasies use and the attitudes towards them. Research usually relies on ad hoc items on the Likert scale that ask about fantasies frequency, arousability, and related emotions [7,27,28,29,30,31,32]. A valuable example is represented by the study of Ellis and Simons [1], who used an ad hoc questionnaire based on evolutionary theory to test some gender differences in the frequency, use, and attitudes towards erotic fantasies. Sue [4] investigated fantasizing activity during sexual intercourse in both genders, using several ad hoc questions. Besides the specific contents of the fantasies, both genders reported that the primary purpose of these fantasies was to enhance their sexual arousal from the beginning of their partnered sexual experiences. More recently, Wu et al. [33] created an ad hoc protocol using a list of erotic fantasies and connected emotions, as theorized by Singer [34], such as feeling excited, satisfied, rested/released, guilty, anxious, frustrated, happy, and embarrassed/ashamed.

Regarding validated measures, a masterful contribution dates to the work on the “Daydreaming inventory for married women” by Hariton and Singer [35]. In their study, the authors explored the variations in erotic fantasies in women and tested the validity of some theoretical models regarding the functioning of these fantasies. This measure specifically explored various levels such as the negative and positive reactions to erotic fantasies, and their acceptance, use, attitudes, and contents. Despite the great contribution to erotic fantasies research, a real adaptation of the tool for the general population in the subsequent nearly 50 years has never been reported, making the tool obsolete today. Giambra & Singer [36] presented the Sexual Daydreaming Scale (SDS), a short tool for assessing the frequency and contents of fantasies with 12 items. However, the SDS does not differentiate between fantasies frequency and contents. In addition to this, it does not explore the attitudes and emotions related to fantasies.

As highlighted by Cartagena-Ramos et al. [37], questionnaires on fantasies are not often applied to sociocultural contexts that are different to those of their authors, with limitations regarding their replicability and reliability. Moreover, most of the measures are restricted to the heterosexual, cisgender population, failing to capture other possible sexual identities, behaviors, and expressions [38,39,40]. Thus, studies testing updated and comprehensive measures and related psychometric properties are urgently needed.

### The Current Study

In line with these considerations, the current study is part of a wider project that aims to analyze the psychometric properties of a composite measure for sexual desire called “Sexual Desire and Erotic Fantasies questionnaire (SDEF)”. The SDEF is divided into three independent measures (1. Sexual Desire; 2. Use of Erotic Fantasies; and 3. Erotic Fantasies Inventory) [41,42] that can be used separated or together for a general overview of the desire function. The creation of the SDEF was driven by the need to have a tool that is able to explore the different aspects of the desire experience, rather than improve the current available measures, especially one that can be used in clinical settings for the investigation of the key components that should be observed in the assessment of sexual dysfunctions, as highlighted by the major diagnostic classifications such as the DSM-5 and ICD11 [43,44].

In this paper, we will test and discuss the results of the validation study on the Sexual Desire and Erotic Fantasies questionnaire—Part 2 Use of erotic fantasies (SDEF2), which is focused on the frequency, use, attitudes, emotions, and communication linked to sexual fantasizing activity. The SDEF2 was created based on the clinical need to detect different manifestations of sexual desire in order to better evaluate a patient’s desire experience. For this reason, the authors have chosen to devise five specific subscales to investigate: a general dimension of the frequency of erotic fantasies in different sexual and non-sexual contexts; a dimension describing the attitudes towards the normativity of phantasmatic activity; a dimension expressing the value given to this phantasmatic activity by an individual; a dimension that gathers the possible negative emotional reactions to the experience of these erotic fantasies; and a dimension expressing the communication and realization of sexual fantasies with partners. This multidimensional outlook is considered to be one of the strengths of the SDEF2, which may help clinicians in their assessment of desire-related difficulties and can be used in research, as well in deepening the specific characteristics of erotic fantasies. 

Furthermore, a sex-positive approach [40] was considered to build the SDEF2 as a tool that is accessible to all individuals, regardless of their gender identities, sexual orientations, relational/romantic status, and sexual behaviors. Specifically, attention was paid to write items with inclusive language, which is more capable of describing the different manifestations of human sexuality (such as non-penetrative sexual behaviors), particularly in the Italian language, in which gender binary declinations can create difficulties and misinterpretation. In this sense, a sex-positive approach recognizes the tremendous cultural diversity of sexual practices, acknowledging a substantial variation in personal meanings and preferences over time and space.

The main aim of the present study was to develop and validate the SDEF2. Specifically, the study focused on testing the internal reliability, as well as the construct and discriminant validity of the questionnaire. Secondly, the study aimed to explore some characteristics of the erotic fantasies’ dimensions assessed by the SDEF2, such as the associations with the sociodemographic variables, sexual functioning, gender, and sexual orientation differences within a group of Italian people.

## 2. Materials and Methods

### 2.1. Participants and Procedures

In total, 1819 (1135 women, 661 men, and 23 other genders) volunteers from the general population participated in the SDEF validation study. People were recruited with a snowball technique, sharing advertisements on institutional websites and social networks (e.g., Facebook, Instagram, and LinkedIn). The web survey was available on the Google.form platform and the data were collected from January 2019 to December 2020. The participants completed an informed consent form before accessing the survey. The administered questionnaire was anonymous, took about 20 min to be completed, and no remuneration was provided. The institutional ethics committee of the Dept. of Dynamic and Clinical Psychology and Health Science, Sapienza University of Rome, Italy consented to the conduct of this study on 9 January 2019.

The inclusion criteria were being at least 18 years old and holding an Italian citizenship. A total of forty-six responses (2.53%) were excluded from the present study because they represented duplicated, falsified, or incomplete records. The final group resulted in 1773 participants (1105 women, 645 men, and 23 other genders). To run an explorative and confirmative factorial analysis, the participants were randomly assigned to two different groups that were balanced for gender, age, and sexual orientation (Table 1). The same group of participants was involved in the validation study of the SDEF1 and SDEF3 [41,42].

### 2.2. Measures

Sociodemographic Questionnaire—The participants completed a brief sociodemographic form to collect general information such as age, gender, sexual orientation, marital and relational status, children, education level, work status, religious and political orientation, and being sexually active (having had sexual activity in their lifetime).

Sexual Desire and Erotic Fantasies questionnaire-Part 2. Erotic Fantasy Use scale—The SDEF2 is a questionnaire that was designed by the authors to measure five domains related to erotic fantasies attitudes and use: (1) fantasies frequency, (2) fantasies normality, (3) the importance given to fantasies, (4) the negative emotions related to the experience of erotic fantasies, and (5) the sharing and experiences of erotic fantasies with regular partners. In the design phase of the measure, the SDEF2 items were constructed by the authors based on a literature revision of the existing measures on sexual fantasies and clinical experience with working on this topic with individuals and couples. In detail, the five areas of interest were identified based on a comparison between the main criteria of desire-related problems in diagnostic classifications [10,43,44,45]. Subsequently, the available measures from the literature were revised and 26 items were developed, fitting with the areas identified. In this process, the authors paid particular attention to the use of inclusive language that could refer desire to any erotic activity, not only penetrative sex (e.g., kissing, body stimulation, oral sex, and masturbation) and tried to be respectful of any gender identity and sexual orientation. Regarding the response options, a 6-point Likert scale was preferred for the frequency items (e.g., “1. Referring to the LAST 6 MONTHS, how often have you had erotic fantasies?”) and a 5-point Likert scale for the items on attitudes (e.g., “6. How NORMAL do you think it is to have erotic fantasies in general?”). The possibility of having a diversified response mode allowed the participants to express themselves with a consistent variability. Some items have unscored solutions, indicated with a special sign to express the inability to answer the question for a specific reason (e.g., “#. I have never had erotic fantasies”). In total, two items (22 and 24) assessed the emotions perceived during the disclosure of erotic fantasies to a partner, with 14 possible emotional states (discomfort, arousal, embarrassment, fear, happiness, vulnerability, strength, closeness, curiosity, guilt, enthusiasm, irritation, nervousness, and activation), to which the participant may have indicated more than one answer. These two items will not be counted for the purposes of this validation, due to the difficulty of comparing them with the style of the other items constituting the SDEF2, but they should be considered as supportive of this assessment of the phantasmatic experience. Higher scores indicate a higher frequency of sexual fantasies/accordance with the items. After the creation of the SDEF2 pool of items, a group of 10 experts in the fields of psychosexology and sexual medicine independently reviewed the content and validated the 26-item version by sending comments and suggestions to the authors. The criteria used by the experts had a content relevance to the area under investigation and a comprehension of the text (items and answers). Once all the comments from the experts had been collected, the authors revised each item, incorporating minor wording changes. The 26 modified items were retained and pilot tested with 20 volunteers to examine the general comprehension of the questionnaire and then administered in the present study to test its psychometric characteristics. The final version that emerged from the current study presents 21 items (see Appendix A).

Sexual Desire Inventory–2 (SDI-2) [46]—The SDI-2 is a 14-item measure used to evaluate two dimensions of sexual desire: dyadic and solitary sexual desire. Higher scores indicate a higher level of sexual desire. The two-dimensional structure presents satisfying psychometric properties also in the Italian version [47], with the Cronbach’s alpha coefficients in this study being equal to 0.88 for dyadic and 0.91 for solitary sexual desire.

International Index of Erectile Function (IIEF) [48]—The IIEF is a widely used 15-item questionnaire for the evaluation of male erectile and sexual function. A general index of sexual function and 5 specific dimensions are calculated: sexual desire, erectile function, orgasmic function, satisfaction with intercourses, and overall satisfaction. Higher scores indicate a better functioning. Psychometric studies have reported a good reliability, validity, and discrimination between sexually dysfunctional and healthy people (clinical cut off score < 26). For this study, the IIEF was worded in a way to be completed by all the cisgender men, regardless of their sexual orientation. The Cronbach’s alpha in this study ranged from 0.87 (orgasmic function) to 0.93 (overall satisfaction).

Female Sexual Function Index (FSFI) [49]—The FSFI is an established 19-item instrument providing information on general sexual functioning and 6 specific dimensions: sexual desire, sexual arousal, lubrication, orgasm, sexual pain, and sexual satisfaction. Higher scores indicate a better functioning. The measure presents good test–retest reliability, internal consistency, validity, and discrimination between sexually dysfunctional and healthy people (clinical cut off score < 26.55), also in the Italian version [50]. For this study, the FSFI was worded in a way to be completed by all the cisgender women, regardless of their sexual orientation. The Cronbach’s alpha in this study ranged from 0.81 (sexual arousal) to 0.92 (sexual pain).

Marlowe–Crowne Social Desirability Scale-Short Form (MCSDS–SF) [51]—The MCSDS–SF is a 13-item measure that was developed as a means of measuring socially desirable responses. Higher scores indicate a higher tendency to respond in a more socially desirable way. The Cronbach’s alpha value for this measure was 0.91. The MCSDS–SF was used as a covariate in the analysis of the current study to limit the effects of social desirability.

### 2.3. Statistical Analytic Strategy

The psychometric properties of the SDEF2 were tested following different procedures. Conceptualized as a formative measure, where latent constructs depend on the operationalization of the sexual desire facets that are strictly dependent on the construction of the items, the construct validity was estimated at the item level with a principal component analysis (PCA). A direct oblimin rotation was used and the number of factors selected was calculated by a parallel analysis, in conjunction with the Guttman–Kaiser criterion, using a Monte Carlo PCA for the parallel analysis by Watkins [52]. After the establishment of a satisfying model, a path diagram was drawn and tested with a confirmatory factor analysis (CFA). The internal consistency was assessed using Cronbach’s alpha. The composite reliability (CR) and average variance extracted (AVE) values were examined. Pearson correlations (2-tailed) and one-way and two-way multivariate analyses of covariance (MANCOVAs) were used to explore the associations between the sexual fantasies use dimensions and sociodemographic variables, sexual functioning, gender, and sexual orientation differences within a group of Italian people. The age, relational status, and social desirability effects were controlled by putting them as covariates in the MANCOVAs. The PCA, Cronbach’s Alpha values, Pearson correlations, and MANCOVAs were performed using IBM SPSS 27.0 and the CFA was tested with IBM SPSS Amos 22 (IBM Corp, Armonk, NY, USA).

## 3. Results

The participants’ mean age was 29.31 ± 10.35 years (range 18–78). Table 1 shows that the sociodemographic variables assessed within the total group of participants reached (n = 1773). A total of two subgroups were randomly extracted to run separately exploratory and confirmatory factorial analyses (Group 1 n = 887; and Group 2 n = 886).

### 3.1. Principal Component Analysis

Group 1 was used to test the factorial structure of the SDEF2 with principal component analyses (PCAs). After excluding the multiple choice non-quantitative items (22 and 24) used to deepen the understanding of the emotional connotation of sharing fantasies with a partner, PCAs were run on the remaining 24 items of the SDEF2 using a direct oblimin rotation. A Kaiser–Meyer–Olkin value of 0.82 supported the adequacy of the sample. The significance of the Bartlett test of sphericity (χ^2^ = 11,836.301; *p* < 0.001) meant that the item correlations were large enough to conduct PCAs. A Monte Carlo Parallel Analysis identified five components, accounting for 63.71% of the total variance. The item selection was based on loadings higher than 0.4 for the respective factors. In total, three items (9, 13, and 14) were loaded below 0.4 for all the factors or were loaded higher than 0.4 for more than one factor. Thus, they were excluded from the following analyses. Table 2 presents the retained 21 items’ component loadings.

### 3.2. Confirmatory Factorial Analysis

To validate the five-factor structure identified by the PCA, a CFA was run on Group 2 measuring the model fit, comparison, and parsimony’s indices. A maximum likelihood estimation method was used. The χ^2^ value for the model (Figure 1) was significant (χ^2^ = 1025.51, *p* < 0.001). The RMSEA was 0.055 (90% CI = 0.052–0.059). The other fit indices that were evaluated included the GFI (0.95), NFI (0.94), and CFI (0.95). A good fit was reached for all the measures except for the χ^2^ value, due to its sensitivity to large sample sizes (n > 200). The regression coefficients for this model ranged from 0.41 to 0.97 and were all statistically significant (*p* < 0.001).

CFAs were also run separately for gender (cisgender women and men) and sexual orientation (heterosexual, bisexual, and homosexual) to test the fits in the subgroups. The fit indices are reported as: cisgender women (RMSEA = 0.054 (90% CI = 0.05–0.059); GFI = 0.94; NFI = 0.94; and CFI = 0.95); cisgender men (RMSEA = 0.061 (90% CI = 0.056–0.067); GFI = 0.93; NFI = 0.91; and CFI = 0.93); heterosexual participants (RMSEA = 0.058 (90% CI = 0.055–0.062); GFI = 0.94; NFI = 0.93; and CFI = 0.94); bisexual participants (RMSEA = 0.052 (90% CI = 0.036–0.067); GFI = 0.91; NFI = 0.89; and CFI = 0.95); and homosexual participants (RMSEA = 0.055 (90% CI = 0.041–0.069); GFI = 0.9; NFI = 0.91; and CFI = 0.94).

### 3.3. Internal Consistency, Convergent, and Discriminant Validity

Based on the total group (n = 1773), the intercorrelations between the five factors were all statistically significant (Table 3), except for F3 with F4. The internal consistency was assessed: the Cronbach α coefficients were satisfactory (F1 = 0.68; F2 = 0.67; F3 = 0.89; F4 = 0.88; and F5 = 0.85); the composite reliability for each construct was above the expected threshold of 0.70 (F1 = 0.73; F2 = 0.8; F3 = 0.91; F4 = 0.91; and F5 = 0.88); and the average variance extracted value for each factor was above the expected threshold of 0.50, except for F1 (F1 = 0.37; F2 = 0.57; F3 = 0.77; F4 = 0.62; and F5 = 0.65). Table 3 also reports the Pearson’s correlations with the SDI-2, FSFI, and IIEF scores to verify the convergent and discriminant validity.

### 3.4. Validity Evidence Based on the Relationship with Other Variables

Focusing on the SDEF2 description, the associations with the sociodemographic variables were explored. Table 4 reports the Pearson’s correlations with age, being in a relationship, education level, and political and religious attitudes, sexual intercourses, and social desirability. The different dimensions of sexual fantasies use were shown to be significantly associated with sociodemographic variables such as age, relationship status, having children, education level, sexual intercourses, and political and religious attitudes. Due to the importance highlighted in the current results and similar constructs in the literature, age, relationship status, and social desirability were considered as covariates in the following analyses, aiming to explore the possible differences in erotic fantasies use among different genders and sexual orientations.

Both to observe the stability of the measure between genders and orientations and to further the gender and sexual orientation debate [38,39] with regard to the phantasmatic experience, which is lacking, poorly adjusted, and biased, it was considered important to investigate any differences or similarities between the five factors investigated by SDEF2 and the different genders and orientations. Due to the limited number of participants reporting “other genders” (transgender/gender-nonconforming), asexual, and pansexual orientations, we decided to focus on people declaring themselves as cisgender women and men (gender) and heterosexual, bisexual, or homosexual (sexual orientation). A two-way MANCOVA using age, being in a relationship, and social desirability as covariates was run to highlight the gender and sexual orientation differences in the SDEF2 factors. Gender and sexual orientation were considered as the independent variables, while the SDEF2 dimensions were used as the dependent ones. The findings are reported in Table 5, showing significant results for gender, sexual orientation, and gender x sexual orientation (See Figure 2).

To explore if the SDEF2 dimensions were able to differentiate between the clinical scores of the FSFI and IIEF, two one-way MANCOVAs using age, being in a relationship, and social desirability as covariates were run to highlight the sexual functioning differences in the SDEF2 factors. The clinical scores of the FSFI for women and the IIEF for men were considered as the independent variables, while the SDEF2 dimensions were used as the dependent ones. The findings are reported in Table 6, showing significant higher scores for all the SDEF2 fantasies dimensions for the participants with FSFI and IIEF functional scores compared to the ones with clinical scores. An exception is reported for “fantasies importance” in women, with no significant differences between the groups. Negative emotions were significantly higher in the clinical groups rather than functional ones. The effect size ranged from small to medium (0.006–0.1).

## 4. Discussion

The current study aimed to develop a self-administered measure of erotic fantasies use and attitudes and evaluate its psychometric properties. A five-factor structure was hypothesized during the development of the SDEF2. PCAs and Monte Carlo Parallel Analyses were performed, confirming the assumed structure. CFAs confirmed a good fit of the SDEF2, and its internal consistency showed satisfying results. The final version included 21 items explaining 63.71% of the variance. The factors highlighted were:F1. Fantasies Frequency (five items)—A dimension describing the self-reported frequency of erotic fantasies in different sexual and non-sexual contexts. Higher scores indicated a higher frequency of erotic fantasies.F2. Fantasies Normality (three items)—A dimension that described how much the person felt that having erotic fantasies in general, during masturbation and during sexual activity with a partner, was “normal”. Higher scores indicated a higher perception of erotic fantasies as a standard and regular expression of the sexual experience.F3. Fantasies Importance (three items)—A dimension gathering the importance and functionality attributed to erotic fantasies within the sexual experience. Higher scores indicated a higher value given to erotic fantasies.F4. Negative Emotions (six items)—A dimension collecting the range of negative effects that might be experienced as a reaction to erotic fantasies, such as discomfort, worry, a sense of guilt, anger, frustration, and embarrassment. Higher scores indicated a higher presence of negative emotions related to erotic fantasies.F5. Sharing and Experiencing (four items)—A dimension describing how often a person shared and experimented with their erotic fantasies with a regular partner (if any). Higher scores indicated a higher level of sharing and practicing sexual fantasies with a regular partner.

The presented factorial structure provides a measure of the frequency of erotic fantasies and thoughts (F1), which is often used in research as an expression of sexual desire and erotophilia/erotophobia [8,21,22,53]. The possibility of describing the different aspects of erotic fantasies attitudes and uses may be advantageous for both deepening the study of fantasies’ role in the sexual response and for their clinical application in sexual and couples therapy [10,45]. Fantasies are an important tool that has been extensively used in therapy to investigate, stimulate, and re-connect partners’ intimacy. However, there is still a wide variability among clinicians in how to assess and use sexual fantasy in treatments [10]. Specifically, the SDEF2 section related to attitudes towards fantasies (F2 and F3) and negative emotions (F4) may favor the acknowledgement of how people interpret fantasies [6]: Do they consider erotic fantasies to be a useful tool for their individual and partnered sexuality? Do they believe that fantasies are something that may distract them from the real sexual act (creating a parallel pleasant/unpleasant experience)? Having a more aware vision of one’s relationship with their fantasies might be useful both for dealing with desire issues and other problems within the sexual and relational sphere [12]. It should be noted that no significant correlation was found between F3 and F4. This may indicate an independence between the importance a person places on their phantasmatic activity for arousal and the sexual experience with the type of negative emotions that might be experienced in association with their phantasies. In any case, both factors are considered important to consider for measurement. Moreover, F5 could be useful for gathering information on whether a couple communicates about intimate aspects such as fantasies, and if they put them into practice. This allows, especially in the clinical setting, for a discussion together on how an individual or a couple sexually communicates about their wishes, boundaries, and how sexuality is negotiated [45].

Regarding the associations with the sociodemographic variables, older age, being in a relationship, having children, and higher levels of education seem to be protective factors for the negative emotions related to erotic fantasies. In this sense, it should be recognized how personal experience and a partners’ presence and support may improve the personal attitudes towards one’s fantasies in a more compassionate way [24,54,55,56]. People expressing more conservative political and religious attitudes seem to report a lower frequency of erotic fantasies and more negative attitudes towards them. These results are in line with previous studies [57,58,59], although they may be influenced by an adherence to stereotypes and/or social desirability, which may significantly affect the self-reported measures within sexuality research [7]. 

In addition, an interesting feature emerges from the associations with the SDI1, the FSFI, and the IIEF. A higher frequency of erotic fantasies and sharing and experiencing them (F1 and F5) were associated with a higher sexual functioning and satisfaction scores in both women and men. Regarding the normality and importance given to fantasizing, more positive attitudes (F2 and F3) were associated with a higher desire in both genders. Moreover, a higher presence of negative emotions related to these erotic fantasies was associated with a lower functioning and satisfaction scores in both genders. These results seem to confirm that frequent and positive fantasizing activity is associated with higher levels of sexual functioning and satisfaction, suggesting, on the one hand, that erotic fantasies could be a fundamental expression of healthy and satisfying sexuality, while, on the other hand, a problem in sexual functioning may negatively influence this fantasizing activity [45,60].

Regarding the gender differences in fantasizing, controlling for the age, relational status, and social desirability effects, men reported a significantly higher frequency of erotic fantasies than women. No differences were found regarding the other SDEF2 domains. In line with other studies [21,33], a small gender difference in erotic thoughts frequency was expected (partial eta^2^ = 0.013), although this reporting may be influenced by gender role expectations.

Considering sexual orientation, the heterosexual participants seemed to report significantly lower scores for fantasies frequency and normalization compared to the bisexual and homosexual participants. Specifically, intersecting for gender and sexual orientation, bisexual women reported a higher frequency compared to other women and higher scores for normalization compared to all the groups. These results are central, as they add data to the scarce literature on bisexuality [38].

Another important result was with regard to the ability of the SDEF2 to differentiate between sexually clinical and functional women and men. Table 6 shows how groups of women and men, respectively, discriminated by the FSFI and IIEF clinical cut-off scores [48,49], achieved significantly different scores in all the areas of the SDEF2 domains (except for F3 in women), with the clinical population reporting lower scores for F1, F2, F3, and F5 and a higher presence of negative emotions for F4. These results seem to suggest the ability of the SDEF2 to discriminate between sexually functional and dysfunctional men and women. Therefore, the SDEF2 could be suggested as a possible screener to provide directions to clinicians in the assessment of sexual difficulties.

The SDEF2 can be considered to be a sex-positive questionnaire [40] that helps to overcome the significant bias present in most of the tools that have been used so far in the literature: the theoretical focus on married and monogamous heterosexual couples, which excludes all those sexual and relational expressions that move away from this heteronormative and dyadic vision of human sexuality [38,39,61,62]. 

The present research has some limitations that should be discussed. (i) The participants were selected with a “snowball” technique; therefore, it is impossible to generalize these results for the Italian population, despite the large number of participants involved, and should be replicated with a randomized sample. (ii) The SDEF2 was created as a tool that measures the personal perception of one’s erotic fantasies. In this sense, the responses can be easily falsified by the respondents. Therefore, any assertion about people’s real fantasizing activity and attitudes should be done with extreme caution. To limit this bias, the study used a large group of participants and a social desirability measure was considered. (iii) The test–retest reliability was not assessed for this study. For this reason, further studies should be conducted to replicate the present findings and extend the psychometric understanding of the SDEF2. Moreover, future studies should consider extending the evaluation of sexual desire to different sexual identities and orientations, behind binarism. Multicultural studies on the SDEF2 psychometric properties and, more in general, on sexual fantasies, to explore the differences and similarities between countries are needed.

## 5. Conclusions

Erotic fantasizing activity is a complex and largely unknown area of investigation, but studies like the present one may help in taking a small step forward. Specifically, the present study extends the current knowledge about the different characteristics of erotic fantasies use and attitudes and their connections with sexual functioning among different genders and sexual orientations. This may be important not only for advances in research, but also for improvements in clinical practice [10,40]. Sexual therapists should acknowledge the role played by erotic fantasies and use specific techniques in their clinical practice to improve sexual functioning, sexual communication, relational intimacy, and satisfaction [10,45]. For this purpose, our results seem to validate the idea that the SDEF2 could be a useful and valid measure for assessing the different expressions and attitudes towards erotic fantasies, and its use should be recommended for clinical and research purposes. Moreover, we suggest assessing the SDEF2 in association with the SDEF1 and SDEF3 [41,42] to provide a more complete view of desire and erotic fantasies.

## Figures and Tables

**Figure 1 healthcare-11-01159-f001:**
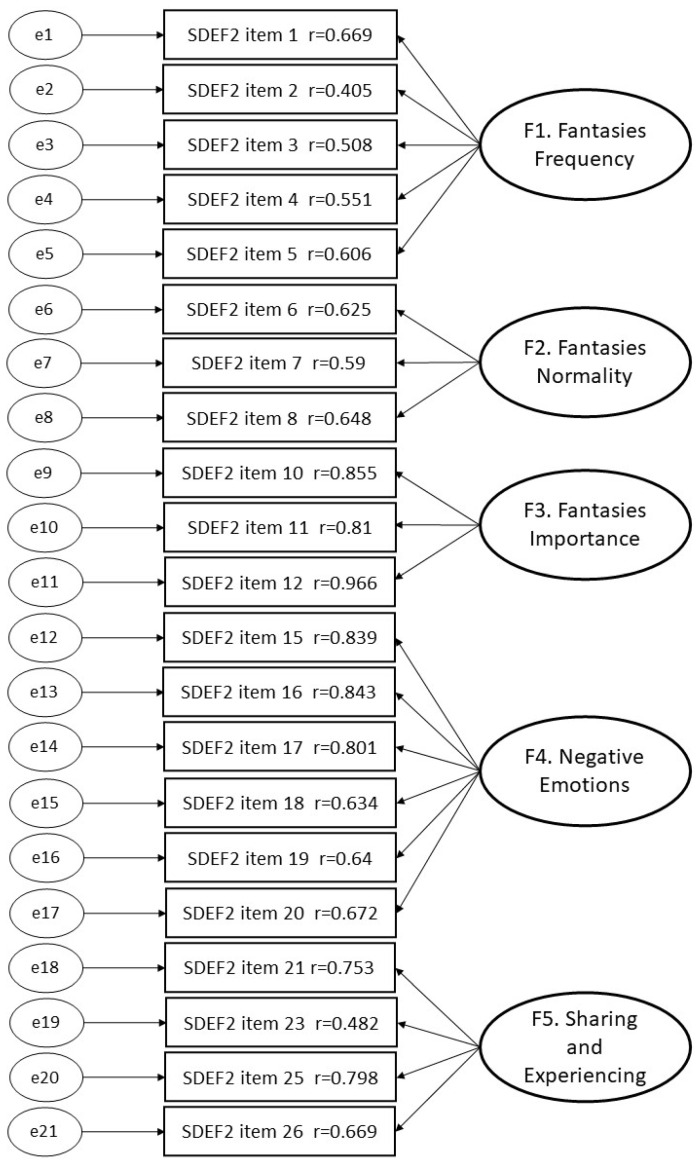
Path diagram of the confirmatory factorial analysis of the SDEF2 (n = 886).

**Figure 2 healthcare-11-01159-f002:**
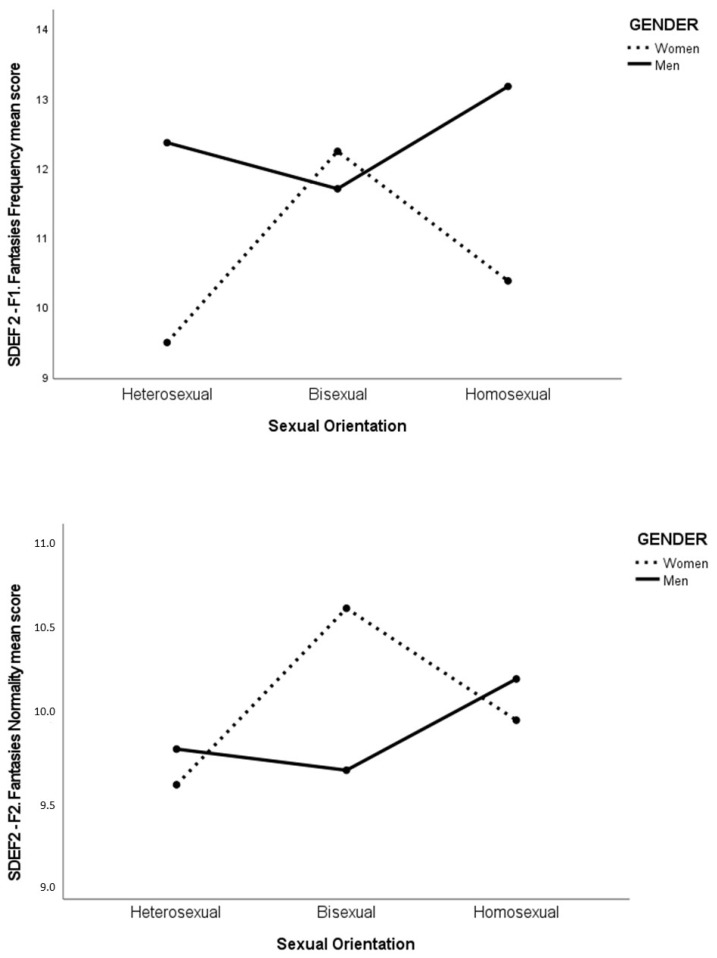
Diagrams of gender x sexual orientation for SDEF1 factors (MANCOVAs) (n = 1729).

**Table 1 healthcare-11-01159-t001:** Sociodemographic variables description.

Variables		Group 1 (n = 887)	Group 2 (n = 886)	Total Group (n = 1773)
		M ± DS (Min–Max)	M ± DS (Min–Max)	M ± DS (Min–Max)
Age		29.3 ± 10.42 (18–78)	29.32 ± 10.28 (18–65)	29.31 ± 10.35 (18–78)
		n (%)	n (%)	n (%)
Gender	Female	555 (62.57)	550 (62.08)	1105 (62.32)
	Male	320 (36.08)	325 (36.68)	645 (36.38)
	Transgender	3 (0.34)	3 (0.34)	6 (0.34)
	Non-binary	9 (1.01)	8 (0.91)	17 (0.96)
Sexual Orientation	Heterosexual	705 (79.48)	703 (79.35)	1408 (79.41)
	Bisexual	80 (9.02)	82 (9.26)	162 (9.14)
	Homosexual	89 (10.03)	89 (10.05)	178 (10.04)
	Asexual	10 (1.13)	9 (1.02)	19 (1.07)
	Pansexual	3 (0.34)	3 (0.34)	6 (0.34)
Marital Status	Unmarried	763 (86.02)	736 (83.07)	1499 (84.55)
	Married	96 (10.82)	125 (14.11)	221 (12.46)
	Separated	24 (2.71)	24 (2.71)	48 (2.71)
	Widowed	4 (0.45)	1 (0.11)	5 (0.28)
Relational Status	Single	333 (37.54)	293 (33.07)	626 (35.31)
	Couple	532 (59.98)	576 (65.01)	1108 (62.49)
	Polyamory	22 (2.48)	17 (1.92)	39 (2.2)
Children	No	787 (88.73)	764 (86.23)	1551 (87.48)
	Yes	100 (11.27)	122 (13.77)	222 (12.52)
Education Level	Middle School	19 (2.14)	21 (2.37)	40 (2.26)
	High School	286 (32.24)	333 (37.58)	619 (34.91)
	University	443 (49.94)	396 (44.7)	839 (47.32)
	PhD and Postgrads courses	139 (15.67)	136 (15.35)	275 (15.51)
Work Status	Student	422 (47.58)	414 (46.73)	836 (47.15)
	Employed	241 (27.17)	274 (30.93)	515 (29.05)
	Freelance	150 (16.91)	140 (15.8)	290 (16.36)
	Unemployed	64 (7.22)	56 (6.32)	120 (6.77)
	Retired	10 (1.13)	2 (0.23)	12 (0.68)
Sexual Intercourse in Life	Never	45 (5.07)	54 (6.09)	99 (5.58)
	Yes	842 (94.93)	832 (93.91)	1674 (94.42)
Sexual Intercourse in the last 6 months	No	138 (15.56)	110 (12.42)	248 (13.99)
	Yes	749 (84.44)	776 (87.58)	1525 (86.01)

**Table 2 healthcare-11-01159-t002:** Principal component analysis matrix (n = 887)—SDEF2’s 21 items extracted from the 26-pilot tested version.

	Factors Extracted				
	F1. Fantasies Frequency	F2. Fantasies Normality	F3. Fantasies Importance	F4. Negative Emotions	F5. Sharing and Experiencing
SDEF2_03	0.727				
SDEF2_01	0.701				
SDEF2_02	0.625				
SDEF2_04	0.471				
SDEF2_05	0.44				
SDEF2_07		0.839			
SDEF2_06		0.829			
SDEF2_08		0.575			
SDEF2_11			0.93		
SDEF2_12			0.899		
SDEF2_10			0.807		
SDEF2_16				0.858	
SDEF2_15				0.829	
SDEF2_17				0.818	
SDEF2_20				0.769	
SDEF2_18				0.729	
SDEF2_19				0.727	
SDEF2_23					0.834
SDEF2_26					0.821
SDEF2_21					0.812
SDEF2_25					0.761
Rotation Method: Direct Oblimin					

**Table 3 healthcare-11-01159-t003:** Person’s correlation matrix between SDEF2 Factors, SDI-2, FSFI, and IIEF (n = 1773).

	SDEF2 F1	SDEF2 F2	SDEF2 F3	SDEF2 F4	SDEF2 F5
SDEF2-F1. Fantasies Frequency	1	0.446 **	0.405 **	0.108 **	0.344 **
SDEF2-F2. Fantasies Normality	0.446 **	1	0.472 **	−0.092 **	0.202 **
SDEF2-F3. Fantasies Importance	0.405 **	0.472 **	1	0.043	0.154 **
SDEF2-F4. Negative Emotions	0.108 **	−0.092 **	0.043	1	−0.073 *
SDEF2-F5. Sharing and Experiencing	0.344 **	0.202 **	0.154 **	−0.073 *	1
SDI-2—Solitary Desire	0.559 **	0.288 **	0.244 **	0.133 **	0.047 ^†^
SDI-2—Dyadic Desire	0.541 **	0.241 **	0.24 **	0.021	0.235 **
FSFI—Sexual Desire	0.466 **	0.151 **	0.138 **	0.015	0.332 **
FSFI—Arousal	0.234 **	0.091 *	0.03	−0.092 *	0.4 **
FSFI—Lubrication	0.201 **	0.089 *	0.038	−0.074 ^†^	0.335 **
FSFI—Orgasm	0.159 **	0.069 ^†^	−0.017	−0.119 **	0.333 **
FSFI—Satisfaction	0.114 **	0.008	−0.019	−0.134 **	0.436 **
FSFI—Pain	0.157 **	0.055	0.014	−0.08 *	0.329 **
FSFI—Total Score	0.237 **	0.084 *	0.026	−0.103 *	0.425 **
IIEF—Sexual Desire	0.378 **	0.142 **	0.141 **	−0.128 *	0.274 **
IIEF—Erectile Function	0.188 **	0.1 ^†^	0.025	−0.232 **	0.431 **
IIEF—Orgasmic Function	0.138 **	0.109 *	0.053	−0.19 **	227 **
IIEF—Intercourse Satisfaction	0.204 **	0.08 ^†^	0.029	−0.227 **	0.488 **
IIEF—General Satisfaction	0.127 *	0.059	0.022	−0.243 **	0.499 **
IIEF—Total Score	0.217 **	0.108 ^†^	0.043	−0.25 **	0.472 **

Note: ^†^ = *p* < 0.05; * = *p* < 0.01; ** *p* < 0.001.

**Table 4 healthcare-11-01159-t004:** Person’s correlation matrix between SDEF2 factors, social desirability (MC-SDS), and sociodemographic variables (n = 1773).

	SDEF2 F1	SDEF2 F2	SDEF2 F3	SDEF2 F4	SDEF2 F5
Age	0.076 *	0.061 †	0.108 **	−0.144 **	−0.026
Being in a Relationship	0.016	−0.008	−0.006	−0.119 **	0.284 **
Having Children	0.025	0.017	0.098 **	−0.122 **	0.001
Education Level	0.027	0.081 *	0.011	−0.081 *	−0.023
Political Conservativisms (Right-winged)	−0.041	−0.145 **	−0.023	−0.029	0.034
Political Involvement	0.114 **	0.128 **	0.036	−0.014	−0.002
Religious Education	−0.039	−0.038	−0.013	0.009	−0.012
Religiousness	−0.098 **	−0.155 **	−0.027	0.055 †	−0.034
Religious Involvement	−0.122 **	−0.157 **	−0.056 †	0.035	−0.061 ^†^
Sexual Intercourse in Life	0.133 **	0.067 *	0.047 †	−0.097 **	0.208 **
Sexual Intercourse in the last 6 months	0.168 **	0.036	0.035	−0.09 **	388 **
Social Desirability (MC-SDS)	−0.117 **	−0.074 *	−0.073 *	−0.227 **	0.016

Note: ^†^ = *p* < 0.05; * = *p* < 0.01; ** *p* < 0.001.

**Table 5 healthcare-11-01159-t005:** MANCOVAs using gender and sexual orientation as independent variables and SDEF2 factors as dependent ones (n = 1729).

	Women (n = 1088) M ± DS	Men (n = 641) M ± DS		Δ	F_(1,1724)_	*p*	95% CI		Partial Eta^2^
							Lower Bound	Upper Bound	
SDEF2-F1. Fantasies Frequency	9.79 ± 4.21	12.52 ± 3.74		2.73	22.98	<0.001	-4.12	−1.47	0.013
SDEF2-F2. Fantasies Normality	9.7 ± 1.96	9.91 ± 1.83		0.21	-	0.334	−0.874	0.394	-
SDEF2-F3. Fantasies Importance	8.5 ± 2.67	8.46 ± 2.47		0.04	-	0.118	−0.842	0.88	-
SDEF2-F4. Negative Emotions	3.51 ± 4.22	3.94 ± 4.5		0.43	-	0.627	−0.597	2.192	-
SDEF2-F5. Sharing and Experiencing	4.5 ± 4.01	4.64 ± 3.84		0.14	-	0.364	−1.805	0.724	-
	**Heterosexual** **(n = 1404)** **M ± DS**	**Bisexual** **(n = 152)** **M ± DS**	**Homosexual** **(n = 173)** **M ± DS**	**Post hoc** **Bonferroni**	**F_(1,1724)_**	** *p* **	**95% CI**		**Partial eta^2^**
							**LOWER BOUND**	**Upper Bound**	
SDEF2-F1. Fantasies Frequency	10.45 ± 4.26	12.16 ± 3.77	12.45 ± 3.93	He < Bi He < Ho	5.71	0.003	1.33	5.337	0.007
SDEF2-F2. Fantasies Normality	9.66 ± 1.98	10.42 ± 1.56	10.18 ± 1.48	He < Bi He < Ho	4.98	0.007	0.224	2.14	0.006
SDEF2-F3. Fantasies Importance	8.4 ± 2.63	9.17 ± 2.3	8.57 ± 2.5		-	0.138	−0.39	2.213	-
SDEF2-F4. Negative Emotions	3.53 ± 4.25	4.7 ± 4.86	3.91 ± 4.33		-	0.082	−3.603	0.614	-
SDEF2-F5. Sharing and Experiencing	4.51 ± 3.95	5.07 ± 4.19	4.46 ± 3.69		-	0.385	−1.143	2.680	-
	**Gender**	**Sexual** **Orientation**	**M**	**SD**	**F_(1,1724)_**	** *p* **	**Partial Eta^2^**		
SDEF2-F1. Fantasies Frequency	Women	Heterosexual	9.44	4.16	9.01	<0.001	0.01		
		Bisexual	12.26	3.89					
		Homosexual	10.42	3.84					
	Men	Heterosexual	12.38	3.77					
		Bisexual	11.82	3.36					
		Homosexual	13.22	3.69					
SDEF2-F2. Fantasies Normality	Women	Heterosexual	9.57	2.01	4.53	0.011	0.005		
		Bisexual	10.6	1.41					
		Homosexual	9.96	1.7					
	Men	Heterosexual	9.83	1.92					
		Bisexual	9.79	1.9					
		Homosexual	10.27	1.38					
SDEF2-F3. Fantasies Importance	Women	Heterosexual	8.4	2.7	-	0.292	-		
		Bisexual	9.3	2.34					
		Homosexual	8.42	2.51					
	Men	Heterosexual	8.4	2.49					
		Bisexual	8.74	2.14					
		Homosexual	8.62	2.51					
SDEF2-F4. Negative Emotions	Women	Heterosexual	3.32	4.07	-	0.148	-		
		Bisexual	4.56	4.72					
		Homosexual	4.62	5.19					
	Men	Heterosexual	3.94	4.55					
		Bisexual	5.18	5.36					
		Homosexual	3.64	4.03					
SDEF2-F5. Sharing and Experiencing	Women	Heterosexual	4.39	3.97	-	0.549	-		
		Bisexual	5.36	4.31					
		Homosexual	4.58	3.6					
	Men	Heterosexual	4.75	3.88					
		Bisexual	4.06	3.626					
		Homosexual	4.41	3.74					

Note: Age, relational status, and social desirability were used as covariates.

**Table 6 healthcare-11-01159-t006:** MANCOVAs having FSFI and IIEF Clinical scores as independent variables and SDEF2 factors as dependent ones (n = 1729).

Women	FSFI Functional Score (n = 647) M ± DS	FSFI Clinical Score (n = 441) M ± DS	Δ	F_(1,1083)_	*p*	95% CI		Partial Eta^2^
						Lower Bound	Upper Bound	
SDEF2-F1. Fantasies Frequency	10.5 ± 4.24	8.75 ± 3.93	1.25	61.83	<0.001	1.618	2.694	0.054
SDEF2-F2. Fantasies Normality	9.79 ± 1.96	9.56 ± 1.97	0.23	6.81	0.009	0.085	0.6	0.006
SDEF2-F3. Fantasies Importance	8.45 ± 2.68	8.58 ± 2.66	0.13	-	0.924	−0.367	0.333	-
SDEF2-F4. Negative Emotions	3.13 ± 3.87	4.07 ± 4.63	0.94	6.92	0.009	−1.243	−0.181	0.006
SDEF2-F5. Sharing and Experiencing	5.79 ± 4.01	2.61 ± 3.16	3.18	119.99	<0.001	2.193	3.15	0.1
**Men**	**IIEF Functional Score** **(n = 527)** **M ± DS**	**IIEF Clinical Score** **(n = 114)** **M ± DS**	**Δ**	**F_(1,636)_**	** *p* **	**95% CI**		**Partial eta^2^**
						**Lower Bound**	**Upper Bound**	
SDEF2-F1. Fantasies Frequency	12.85 ± 3.69	10.96 ± 3.61	1.89	17.11	<0.001	0.896	2.517	0.026
SDEF2-F2. Fantasies Normality	10.04 ± 1.75	9.32 ± 2.08	0.72	12.91	<0.001	0.334	1.14	0.02
SDEF2-F3. Fantasies Importance	9.91 ± 1.83	8.57 ± 2.45	1.34	4.5	0.034	0.043	1.127	0.007
SDEF2-F4. Negative Emotions	3.48 ± 4.09	6.11 ± 5.59	2.63	19.64	<0.001	−3.122	−1.205	0.03
SDEF2-F5. Sharing and Experiencing	5.25 ± 3.75	1.82 ± 2.88	3.43	49.93	<0.001	2.066	3.657	0.073

Note: Age, relational status, and social desirability were used as covariates.

## Data Availability

Data are unavailable due to privacy or ethical restrictions. Any further request can be done to the corresponding author.

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
