# Peer review of "Sexual Desire and Erotic Fantasies Questionnaire: The Development and Validation of the Erotic Fantasy Use Scale (SDEF2) on Experience, Attitudes, and Sharing Issues"

_healthcare, 2023, doi:10.3390/healthcare11081159_

Round 1

Reviewer 1 Report

Please add the chosen method for preparing the questionnaire and the steps related to it in the method section

Why the concept analysis or a qualitative study not used to design the questionnaire?

I think If the questionnaire was designed separately according to gender or some specialized questions were separated according to gender, it would not be more practical

Author Response

1.Please add the chosen method for preparing the questionnaire and the steps related to it in the method section

Response: thank you so much for this comment. The method and the procedure is reported in detail in the method section (under the subparagraph measures, on the description of the SDEF2). We have tried to improve the description of all the phases. We hope that the current version could be clearer for the reader.

2.Why the concept analysis or a qualitative study not used to design the questionnaire?

Response: thank you for this comment. We opted first for a reviewing process of the literature (in particular the measures used and described in the introduction) in order to overcome some of the limitations of previous tools. This process was integrated with a deep discussion of the clinical needs to have a tool that could facilitate the assessment of erotic fantasies with patients. This is because it is an issue that is often a challenge both for patients and for clinicians themselves. We decided not to adopt a qualitative study because we have strong examples in the literature of such studies such as Newbury et al., 2012. We therefore made use of the evidence already available rather than generating new evidence.

3.I think If the questionnaire was designed separately according to gender or some specialized questions were separated according to gender, it would not be more practical

Response: Thank you for this comment. Perhaps we did not quite understand what the reviewer was pointing out. The instrument was designed as 'unisex', in the sense that it does not have any gender-specific declinations, orientation, etc. On the contrary, as is explained in detail in the description of the tool and as can be seen from the tool itself in Appendix 1, we paid particular attention to the use of inclusive language that could refer desire to any erotic activity, not only penetrative sex (e.g., kissing, body stimulation, oral sex, masturbation) and trying to be respectful of any gender identity and sexual orientation.

Reviewer 2 Report

Using a non-probability sample of 1,773 Italian adults, this study evaluated the psychometric properties of the Sexual Desire and Erotic Fantasies questionnaire- Part 2 (SDEF2). My comments are included below.

1.     It was strange to randomly split the sample into two groups, which could not improve the quality of data: a non-probability sample based on a snowball sampling technique (qualitative and non-random sampling method). Use the entire sample!

2.     The analytical approach was unnecessarily complicated. Please consider the following recommendations:

(1)   Present a correlation matrix that includes all the variables in this study (use composite measures, not individual items).

(2)   Use past research and exploratory factor analysis (EFA) to identify dimensions of SDEF2.

(3)   Use confirmatory factor analysis (CFA) with latent variables to assess the goodness-of-fit of the 5-factor model. Make sure to report important goodness-of-fit indices that are typically reported by scholars using the structural equation modeling (SEM) technique. This is to establish factorial and construct validity.

(4)   Conduct multi-group comparisons of the CFA model across gender, sexual orientation, or any other characteristics the authors are interested in this study.

(5)   Use regression models or MACOVA models to link the five dimensions to SDI-2, IIEF, and FSFI to establish criterion and discriminant validity.

3.     Please clearly describe the purpose to examine variations, or lack of, in gender and sexual orientation in MACOVA.

Author Response

  1. It was strange to randomly split the sample into two groups, which could not improve the quality of data: a non-probability sample based on a snowball sampling technique (qualitative and non-random sampling method). Use the entire sample!

Response: Thank you for this comment. The splitting procedure is commonly reported in literature in order to avoid the possibility of running PCA/EFA and CFA on the same group, which will result in an overfitting of the data (as well as being theoretically inconsistent). Many authors suggest to collect a group and randomly splitting it into 2 consistent group which are comparable for some variables of interest (e.g, main sociodemographic) For an update and a discussion of the literature we refer to this recent article : https://link.springer.com/article/10.3758/s13428-021-01750-y

  1. The analytical approach was unnecessarily complicated. Please consider the following recommendations:

(1)   Present a correlation matrix that includes all the variables in this study (use composite measures, not individual items).

(2)   Use past research and exploratory factor analysis (EFA) to identify dimensions of SDEF2.

(3)   Use confirmatory factor analysis (CFA) with latent variables to assess the goodness-of-fit of the 5-factor model. Make sure to report important goodness-of-fit indices that are typically reported by scholars using the structural equation modeling (SEM) technique. This is to establish factorial and construct validity.

(4)   Conduct multi-group comparisons of the CFA model across gender, sexual orientation, or any other characteristics the authors are interested in this study.

(5)   Use regression models or MACOVA models to link the five dimensions to SDI-2, IIEF, and FSFI to establish criterion and discriminant validity.

Response: Thank you for this comment and punctual suggestion. Going in order, (1) correlation matrix with all the measures involved are reported in table 3 and 4. We have presented them in two tables to make them easier to read, but if the reviewer considers it important, we can combine the tables. (2) During the decision between PCA and EFA we considered both the work of Gouvernet et al. 2016 and the one of Coltman et al. 2008. The way the SDEF has been conceptualized, we believe that it can be referred more to a formative measure, because it starts from the operalization of abstract concepts that do not provide a latent variable from which to start, and so it depends directly on item construction. In this sense, PCA (with all its limitations) seems to us the most appropriate solution. We have included the following statement in the article: “Conceptualized as a formative measure, where latent constructs depend on the operationalization of sexual desire facets that are strictly dependent on the construction of the items, construct validity was estimated at item level with Principal Component Analysis (PCA).” (3) Thank you for this point, we have reported the main indices as the χ2 value, RMSEA, GFI, NFI, and CFI. (4) We have added the CFA values for genders and sexual orientation as suggested. (5) Mancovas is reported for IIEF and FSFI, correlations are reported for all the other dimensions.

  1. Please clearly describe the purpose to examine variations, or lack of, in gender and sexual orientation in MACOVA.

Response: Thank you for this comment. Delving into aspects of gender and orientation has a dual function for us. As we have made explicit in the text following your suggestion, “both to observe the stability of the measure between genders and orientations and to further the gender and sexual orientation debate (Nimbi et al., 2020ab) on the phantasmatic experience, which is lacking, poorly adjusted and biased, it was considered important to investigate any differences or similarities in the five factors investigated by SDEF2 and between genders and orientations.

Reviewer 3 Report

Nimbi et al have developed a Sexual Desire and Erotic Fantasies questionnaire by validating the Erotic Fantasy Use scale (SDEF2) among 1,773 Italian participants. I find the manuscript is very interesting, but I have a few suggestions or comments as follows:

Materials and Methods:

I’d define the sexually clinical and functional women and men.

I’d provide the scoring scheme for the 5 and 6- Likert scale

I’d provide the time spent for the respondents to complete the questionnaire.

Results:

About Carlo Parallel Analysis, 5 components were identified and accounted for 63.71% of the total variance. I could not find this in any figures or tables.

 Discussion:

I’d discuss on F3 and F4 as the intercorrelations between 5 factors are not significant. What would be the implications and why do you still keep these factors.

Author Response

REVIEWER 3

Materials and Methods:

I’d define the sexually clinical and functional women and men.

Response: Thank you for this comment. We have added the specific in the abstract. The explanation in detail is reported in the article (both methods section and results),

I’d provide the scoring scheme for the 5 and 6- Likert scale

Response: Thank you for this comment. We have added some examples in the text and the schema is reported in the appendix 1.

I’d provide the time spent for the respondents to complete the questionnaire.

Response: Thank you for this comment. We have added the specific in the methods section (20 minutes).

Results:

About Carlo Parallel Analysis, 5 components were identified and accounted for 63.71% of the total variance. I could not find this in any figures or tables.

Response: Thank you for this comment. As stated in the text, Monte Carlo Parallel Analysis identified 5 components accounting for 63.71% of the total variance. We opted for not repeat this information in table 2 to not be redundant, but if the reviewer prefer to report it also on the table, we can add  it as a note.

 Discussion:

 I’d discuss on F3 and F4 as the intercorrelations between 5 factors are not significant. What would be the implications and why do you still keep these factors.

Response: Thank you for this suggestion. We have added a brief discussion as follows “It should be noted that no significant correlation was found between F3 and F4. This may indicate an independence between the importance a person places on his or her phantasmatic activity for arousal and sexual experience with the type of negative emotions that might be experienced in association with the phantasies. in any case, both factors are considered important to consider for measurement.”

Round 2

Reviewer 2 Report

In the first round of review, I misread why the authors split their dataset into 2 groups. After rereading the manuscript, I believe the authors did this correctly; that is, conducting an EFA using the first group and a CFA using the second group.

However, I continue to be perplexed by this study that used a non-random, thus unrepresentative sample but claimed statistical significance as if the sample was drawn randomly. This flaw is fatal, it cannot be overcome by any sophisticated statistical methods and analyses. Please tune down such claims.

Author Response

Dear reviewer, 

thank you so much for your effort.

We have reviewed the paper accordingly and we have try to improve the "limits" section in order to stress that the results we found cannot be generalized.

Here the part: 

"The present research has some limitations that should be discussed. (i) Participants were selected with a “snowball” technique; therefore, it is impossible to generalize the results for the Italian population despite the large number of participants involved and should be replicated in a randomized sample. (ii) The SDEF2 was created as a tool that measures the personal perception of one's erotic fantasies. In this sense, the responses can be easily falsified by respondents. Therefore, any assertion on people's real fantasizing activity and attitudes should be done with extreme caution. To limit this bias, the study used a large group of participants, and a social desirability measure was considered. (iii) Test-retest reliability was not assessed in this study. For that reason, further studies should be conducted to replicate the present findings and extend the psychometric understanding of the SDEF2. Moreover, future studies should consider extending the evaluation of sexual desire to different sexual identities and orientations, behind binarism. Multicultural studies on the SDEF2 psychometric properties and, more in general, on sexual fantasies to explore differences and similarities between countries are needed."

In this type of studies, it is very common not to have a randomised sampling for the preliminary validation of an instrument. We are sorry if it appeared that we wanted to state the opposite, and we are completely willing to further edit the manuscript if the reviewer deems it useful.
